# Effect of Practice Environment on Nurse Reported Quality and Patient Safety: The Mediation Role of Person-Centeredness

**DOI:** 10.3390/healthcare9111578

**Published:** 2021-11-18

**Authors:** Mu’taman Jarrar, Mohammad Al-Bsheish, Badr K. Aldhmadi, Waleed Albaker, Ahmed Meri, Mohammed Dauwed, Mohd Sobri Minai

**Affiliations:** 1Vice Deanship for Quality and Development, College of Medicine, Imam Abdulrahman Bin Faisal University, Dammam 34212, Saudi Arabia; 2Healthcare Administration Department, Batterjee Medical College, Jeddah 21442, Saudi Arabia; mohammed.ghandour@bmc.edu.sa; 3Department of Health Management, College of Public Health and Health Informatics, University of Ha’il, Ha’il 81451, Saudi Arabia; b.aldhmadi@uoh.edu.sa; 4Department of Internal Medicine/Endocrinology, College of Medicine, Imam Abdulrahman Bin Faisal University, Dammam 34212, Saudi Arabia; wialbakr@iau.edu.sa; 5Department of Medical Instrumentation Techniques Engineering, Al-Hussain University College, Karbala 56001, Iraq; ahmedmeri@hotmail.com; 6Department of Medical Instrumentation Techniques Engineering, Dijlah University College, Baghdad 10022, Iraq; mohalyasari@gmail.com; 7Department of Computer Science, College of Science, University of Baghdad, Baghdad 10070, Iraq; 8College of Business, Universiti Utara Malaysia, Kedah 06010, Malaysia; briminai@gmail.com or

**Keywords:** work environment, quality, patient safety, person-centeredness, workload

## Abstract

This study aims to explore the potential mediation role of person-centeredness between the effects of the work environment and nurse reported quality and patient safety. A quantitative cross-sectional survey collected data from 1055 nurses, working in medical and surgical units, in twelve Malaysian private hospitals. The data collection used structured questionnaires. The Hayes macro explored the mediation effect of person-centeredness between the associations of work environment dimensions and care outcomes, controlling nurses’ demographics and practice characteristics. A total of 652 nurses responded completely to the survey (61.8% response rate). About 47.7% of nurses worked 7-h shifts, and 37.0% were assigned more than 15 patients. Higher workload was associated with unfavorable outcomes. Nurses working in 12-h shifts reported a lower work environment rating (3.46 ± 0.41, *p* < 0.01) and person-centered care (3.55 ± 0.35, *p* < 0.01). Nurses assigned to more than 15 patients were less likely to report a favorable practice environment (3.53 ± 0.41, *p* < 0.05), perceived lower person-centered care (3.61 ± 0.36, *p* < 0.01), and rated lower patient safety (3.54 ± 0.62, *p* < 0.05). Person-centeredness mediates the effect of nurse work environment dimensions on quality and patient safety. Medical and surgical nurses, working in a healthy environment, had a high level of person-centeredness, which, in turn, positively affected the reported outcomes. The function of person-centeredness was to complement the effects of the nurse work environment on care outcomes. Improving the nurse work environment (task-oriented) with a high level of person-centeredness (patient-oriented) was a mechanism through which future initiatives could improve nursing care and prevent patient harm.

## 1. Introduction

The provision of high-quality and safe patient care is a complex process. Nurses in the medical and surgical units frequently experience uncertain work practices that adversely affect their practice and patient care. Work practices in these units are numerous and challenging, and nurses are working in a complex work environment and dealing with patients with a variety of physical and psychological demands. Hence, providing satisfactory care for them is a significant concern. In their report *To Err is Human,* the problem was not bad people in health care, but good people were working in inadequate systems that needed to be made safer [1]. Research in human factors and ergonomics has been increased significantly for greater understanding of human interactions with work systems for reforms and improving care processes [2,3,4,5]. Accordingly, several reforms have been conducted to decrease adverse events and provide safe, patient-centered, timely, effective, efficient, and equitable services in hospitals [6,7]. However, statistics concerning preventable adverse events and deaths remain substantial [8].

Adverse events are an international concern. For example, in the United States, adverse events are the third leading cause of death [9]. Evidence from other countries, such as Malaysia, showed substantial complaints from patients and their families related to adverse events. Thus, the government has prioritized strategic plans to ameliorate nursing care delivery and prevent harmful practices in their healthcare system [10]. Nonetheless, recent studies from Malaysia have revealed that nurses still reported substantial adverse events in private hospitals, specifically in medical and surgical units, as the result of working long shifts [11], a nursing shortage [12], and poor communication with the patients [13].

The global health agenda of the next decade is to provide universal coverage with high-quality, safe, and person-centered care [14]. Researchers, academicians, and policymakers have advocated for this mission, and several studies have been conducted to investigate the determinants of quality care and patient safety as ultimate outcomes of healthcare systems. Among these determinants are nurses’ practice environments [14,15,16]. In the context of a healthcare organization, Lake defined a professional practice environment as “the organizational characteristics of a work setting that facilitate or constrain professional nursing practice” [17].

The Practice Environment Scale of the Nursing Work Index (PES-NWI) recognizes a professional practice environment as comprising five main dimensions: (1) nurse participation in hospital affairs, (2) nurse foundation for quality of care, (3) nurse manager’s ability and leadership support, (4) staffing and resource adequacy, and (5) nurse-physician relationship [17]. Previous studies have recognized the link between these dimensions, and nurses reported outcomes and revealed inconsistent associations. For instance, nurses’ practice environment positively related to the quality of care but was not related to patient safety metrics, such as nosocomial infections, medication errors, patient and family complaints, and pressure ulcers [18]. Similarly, one study found a significant relationship between the nurses’ work environment and reported patient outcomes, while another study found insignificant associations [19,20]. Swiger et al. and Warshawsky and Havens, in their meta-analyses, revealed that nurse work environment dimensions were inconsistently related to the outcomes of care [21,22]. This inconsistency shows the importance of investigating the effect of an intervening factor between the associations of work environment dimensions on care outcomes. A mediator might more clearly explain these inconsistencies [23].

In the last decade, a growing literature has explored the relationships of the work practices environment and nurse reported quality and patient safety. Interestingly, most of this literature has explored the direct association of practice work environment and nurses’ reported outcomes. For instance, nurses who reported higher perceived quality and safety worked in a positive practice environment [14,15,16]. Earlier studies found that nurses working in a healthy practice environment had lower burnout and reported greater patient satisfaction [24]. Ineffective nurse-patient communication might lead to improper care delivery. Nurses in a healthy environment were more likely to integrate patient preferences [25]. Despite this evidence, little is known concerning the role of incorporating patient preferences as a mechanism through which the practice environment affects nurses’ reported quality and safety. Based on this, the current study assumes that the nursing work environment and integrating patient preferences were critical to avoid patient harm and provide high quality and safer care.

Theoretically, this study proposed a model based on the Donabedian Theory for healthcare quality and incorporating the Systems Engineering Initiative for Patient Safety (SEIPS) model [3,26]. The study variables refer to structural, process, and outcome quality (See Figure 1). Donabedian Theory for healthcare quality supposes that structural input with processes leads to outstanding outcome quality. The SEPIS model expands these components by considering human interactions, with a work system, for improving patient safety. This integration of these prominent frameworks in a healthcare quality and work system expands the model’s applicability to the healthcare community [27].

In addition to these theories, introducing clinical and experiential quality concepts was required for a better understanding of human interactions with a work system. Clinical quality refers to task focus quality (for example, adherence to guidelines, staff performance, and minimizing variation), and these refer to structural quality. In contrast, experiential quality refers to the patient-oriented quality to achieve patients’ preferences [28], which matches the process quality. In this study, the role of hospitals and nurse managers in maintaining a healthy nurse work environment refers to the clinical quality (task-oriented), while person-centeredness refers to the experiential quality (patient-oriented). Balancing between task-oriented and patient-oriented to integrate patients’ preferences can optimize the outcomes of care. Thus, the assumption was that nurses working in a favorable environment were more likely to integrate patient preferences, which are of considerable importance in meeting patients’ demands. Based on this assumption, person-centered care, as a core of the caring process, can serve as a mediator between structure quality as work environment and two primary outcomes (i.e., quality care and patient safety).

Person-centered care refers to engaging patients and their families in the process of care by providing effective communication and education of the treatment consequences and nutritional guide [29]. Person-centeredness can help guide efforts and optimize care outcomes [30,31,32]. A patient involved in, and participating in, the treatment processes is less likely to be exposed to adverse events [30].

Few studies have used person-centeredness as a mediator [11,12]. These studies have found that person-centeredness suppresses the impact of workload and working longer shifts on care outcomes. Thus, the mediation role is necessary to better understand the linkage mechanisms [23]. This is the first study that reports nurses’ ratings of quality and safety in medical and surgical units in the Malaysian context. Moreover, it is the first study to explore person-centeredness as a mediator between the relationships of nurses’ work environment dimensions, nurses’ reported quality, and patient safety. The study model provides an insight into the role of maintaining a healthy work environment, in supporting person-centeredness, as human factors for improving quality and patient safety in Malaysia.

## 2. Method

### 2.1. Design and Setting

A cross-sectional study explored the potential mediating effect of person-centeredness between the relationship of nurse practice environment and nurse reported quality and patient safety. The study respondents were licensed nurses registered under the Malaysian Ministry of Health (MOH) in the medical and surgical units in Malaysian private hospitals. Malaysia is a constitutional monarchy federal country located in Southeast Asia with 14 states. The MOH operates the healthcare system in Malaysia, consisting of public and private hospitals. Hospitals in the private sector in Malaysia are part of international health systems for the provision of high-quality care for patients and citizens.

Private hospitals were the study setting because of substantial adverse events that have been reported [33], and Malaysian private hospitals have reported increased medical and legal complaints [13]; hence, the selection of nurses in the medical and surgical units as the study sample was because they are frontline staff who report adverse events and patient and family complaints. They are dealing with alert patients and case complexity. Moreover, nurses in these wards deliver multidisciplinary care such as gastroenterology, cardiology, oncology, nephrology, urology, orthopedics, and ENT [34]. Nurses were chosen as respondents as they are more likely to implement interpersonal interventions as human factors, while physicians mainly implement technical interventions for improving healthcare outcomes [35].

### 2.2. Sample

This research used multi-stage stratified random sampling. This technique offered the opportunity for nurses from various states, hospital sizes, and shifts to be part of the sample. The criterion for hospital inclusion was those hospitals recognized by the Malaysian Association of Private Hospitals. Of the fourteen Malaysian states, only Perlis and Terengganu states have no private hospitals. Hospitals were stratified based on their respective sizes (hospital beds; small < 100, medium 100–200, and large > 200) in the remaining 12 states (Kelantan, Negeri Sembilan, Johor, Malacca, Kedah, Pahang, Perak, Pulau Penang, Sabah, Sarawak, Kuala Lumpur, and Selangor). A total of 10 hospitals were chosen from each stratum using simple random sampling. Stratifying a sample of organizations based on the size will ensure that each one had an equal chance of being selected. As a result, hospitals were stratified based on hospital size from small to large hospitals. Furthermore, all licensed nurses from all working shifts were invited to be part of this study.

### 2.3. Variables and Measurement

Medical and surgical ward nurses in twelve hospitals, in twelve Malaysian States, participated in the study to examine the mediation effect of person-centeredness between the relationships of practice environment dimensions and nurses reported quality and adverse events. Incident reports, patient records, and the Global Trigger tool, developed by Institute of Healthcare Improvement, are the most frequent instruments for tracking and measuring adverse occurrences in hospitals. In hospitals, however, only 10 to 20 percent of adverse events are documented, and 90 to 95 percent of those do not result in patient impact [13]. Furthermore, the hospitals that took part in the study did not grant access to their information systems or patient records. In order to determine the frequency of adverse occurrences in hospitals and measuring the study variables, a self-administered questionnaire was employed. The study variables were structure, process, and outcome quality, as with the following:

#### 2.3.1. Structural Quality

Structural quality reflects the care setting features, such as resources (human resources, materials, and facilities), staff qualifications, and organizational governance [36]. In this study, structural quality refers to the work environment dimensions. The Practice Environment Scale of the Nursing Work Index (PES-NWI), which is validated internationally, measured work environment dimensions [31,37]. The PES-NWI includes five dimensions: nurse participation in hospital affairs, nurse foundation for quality of care, nurse manager’s ability and leadership support, staffing, resource adequacy, and nurse-physician relationship [17,37]. The fourth subscale was considered separately in this study. The degree of agreement of respondents to statements in the work environment subscales was obtained using a 5-point Likert-type rating scale ranging from 1 (strongly disagree) to 5 (strongly agree) to maintain a consistency of measurement across all study variables. The higher the mean, the more preferable the work environment.

#### 2.3.2. Process Quality

Process quality refers to the actual treatment process during the hospitalization [36]. Process quality in the study includes person-centeredness, as the patient is the central point of the caring process. Scholars use the person-centeredness concept to consider a patient’s needs, expectations, and preferences [38] and put the patient’s interest ahead to ensure care is delivered based on these needs [39]. The tool that the Planetree and Picker Institute developed measured this construct [40]. A self-assessment tool on the nurses’ degree of person-centeredness was adapted. The Agency of Healthcare Research and Quality (AHRQ) first developed this tool to assess the Consumer Assessment of Healthcare Providers and Systems (CAHPS) for hospital surveys [31]. The Planetree and Picker Institute labeled eight dimensions of person-centeredness: (1) patient-centered continuity of care, (2) patient-centered documentation and access to the information, (3) patient-centered communication and education, (4) family involvement, (5) personalization of care, (6) environment of care, (7) spirituality, (8) and integrative medicine. The degree of agreement of respondents to statements in the person-centeredness dimensions was obtained using a 5-point Likert-type rating scale, which, in turn, helped render the data of the study comparable to the PES-NWI subscale. The higher mean refers to a higher degree of person-centered care.

#### 2.3.3. Outcome Quality

Outcome quality refers to the consequences of the treatment processes and procedures [41]. It is the dependent variable of the study and includes two dimensions: quality and patient safety. Quality refers to the increase in desired care outcomes consistent with evidence care practices [42]. Nurses were asked to provide a rating on a 5-point Likert-type scale about the degree of agreement of the quality of nursing care in the current ward, last working shift, last year, whether they recommended the services provided by the hospital to their relatives, and whether they recommended the hospital as a good place to work. International nursing literature validates that nurses reported quality of care was close to independent data [34,37].

Patient safety is preventing patient harm or hospital-acquired conditions [43]. Hospital-acquired conditions refer to the negative consequences related to hospitalization [30]. Per the present literature, hospital-acquired conditions in the medical and surgical wards refer to medication errors, bed-sores, falls, nosocomial infections, and patient complaints [44]. Nurses were asked to rate the overall safety on the scale that AHRQ developed and rate the frequency of these acquired conditions. In six items, they were asked to provide their rating about whether they came across these adverse events in their respective units and rate the perceived overall patient safety on a 5-point Likert-type scale. Hence, international literature validates nurse-reported patient safety [37].

### 2.4. Covariates

Nurses’ characteristics, such as nationality, age, gender, level of education, race, marital status, total years of experience, and experience in the current ward served as controls in the data analysis. Additionally, the study accounted for practice characteristics, including hospital size, accreditation and teaching status of hospitals, their total working hours in the last working shift, and the total number of patients under their care as controls in the study. Previous research used these characteristics as control variables with a significant relation to care outcomes in previous research [13].

### 2.5. Ethical Considerations

Appropriate ethical approvals were obtained from the ethical committee of hospitals that agreed to participate in the research. Furthermore, approvals from the original authors were granted to use the study instruments, including Lake to use PES-NWI for the nurse practice environment and the Planetree and Picker Institute for person-centeredness. The informed written consent forms were obtained from all nurses, which randomly selected from a variety of working shifts voluntarily participated in this study.

### 2.6. Data Collection

After ethical approval was received from the hospitals, data were collected between January and March 2015. The data collection used structured questionnaires. The corresponding author visited nursing departments and nurse managers of medical and surgical wards in twelve hospitals and then explained the study’s aim and methods. Participation in the survey was voluntary, and the respondents provided their informed consent. They were assured that data were only for research purposes and would be confidential. The survey took place for two or three times at a place depending on the work shift available in the hospital. The managers have assisted the survey, in distributing and explaining the necessary actions to be done during answering the questionnaires. There were cases, where the managers of the hospital willingly to receive the questionnaires and did the data collection in-house, the researcher just came back to receive the questionnaires. All participants returned the survey in a sealed envelope. The survey collected data from 1055 nurses working in medical and surgical units.

### 2.7. Validity and Reliability

Factor analysis and Cronbach’s alpha coefficient assessed the validity and reliability of the variables. Factor analysis was assessed by Kaiser–Meyer–Olkin (KMO) value and loading factor of the study items, and items with a factor loading of more than 0.50 were retained for further data analysis. The PES-NWI composite mean value was 3.57 with KMO value = 0.88, and Cronbach’s alpha = 0.87 and were more than the recommended value of 0.60 for validity and reliability [45,46]. Cronbach’s alpha results of the elements of the work environment were 0.82 for nurse participation in hospital affairs, 0.73 for nurse foundation for quality, 0.79 for nurse manager’s ability, leadership, and support, and 0.67 for the nurse-physician relationship with mean scores of 3.32, 3.78, 3.68, and 3.64, respectively. The person-centered care composite mean value was 3.65 with KMO = 0.91 and Cronbach’s alpha = 0.89. Additionally, the quality of care mean was 3.73, and the patient safety mean was 3.58, with KMO = 0.88 and Cronbach’s alpha = 0.87, indicating a valid and good internal consistency.

### 2.8. Data Analysis

The data analysis used the Statistical Product and Service Solution (SPSS) software version 21 (SPSS Inc., Chicago, IL, USA). Descriptive statistics illustrated the demographics and practice characteristics of nurses. Analyses of the t-tests and one-way ANOVA identified the relationship of nurses’ work environment, perceived person-centeredness, nurses’ rated quality, and patient safety to their demographic and practice characteristics.

A variety of methods examined the mediation effects. This study used the SPSS Hayes macro method to examine the mediation effect of person-centeredness, as it provides multiple and straightforward mediations to explore the indirect processes [47]. Furthermore, it is a higher power than either the Sobel or Baron and Kenny approaches [48,49,50]. Sobel’s approach considers the indirect effect as normally distributed, whereas Baron and Kenny’s approach does not measure the size of the indirect effect [51]. However, the Hayes macro analysis overcomes these weaknesses by bootstrapping to measure the size of the indirect effect a x b [48,49,50]. A 5000 bootstrap sample and 95% confidence interval (CI) examined the mediation effect of patient-centeredness [48,49,50]. The SPSS Hayes macro analysis provides the confidence interval of the indirect effect a x b, and if the interval does not straddle zero, then the mediation is confirmed [48,49,50]. In addition, the direct effect c′ reports whether the mediator is either fully or partially mediated. The coefficient of path a, path b, and path c′ are reported to identify the exact type of mediation. Based on the decision tree, the similar signs of a x b x c′ paths indicate that the mediator is a complementary mediator [50]. In contrast, the different signs of the coefficients refer to a suppressive or competitive mediator. Moreover, in multidimensional constructs, the evidence of at least one indirect effect is different than zero (confidence interval does not include zero), supporting the conclusion that the proposed mediator significantly mediates the effect of the antecedent variables on the dependent variables [48].

## 3. Findings

### 3.1. Response Rate

Of the 1055 registered nurses in the medical and surgical wards in the 12 participating hospitals, a total of 652 nurses responded completely to the survey, representing a 61.8% response rate. Responses were checked for outliers, and before the main analysis, outliers must be eliminated [45]. At the 0.001 significance level and 84 degrees of freedom (number of items in the model), the chi-square critical value of the study was 129.80. As such, the data analysis results reflect the population of nurses working in medical and surgical wards and are not influenced by the extreme subjects that are not representative of the population [52]. Thus, 69 (10.6%) respondents were deleted from the study; therefore, 583 (89.4%) surveys were considered for further data analysis.

### 3.2. Nurses’ Demographics

The demographic characteristics (see Table 1) indicated that most responding nurses were Malaysian (99.0%) and female (97.6%). Most respondents were Malay (60.0%) and Chinese (21.6%), and the ages of nurses mostly ranged between 25 and30 (43.7%), with a diploma in nursing (84.6%). Further, 65.9% of the respondents had less than five years of working experience; 18.5% had 6–10 years; 7.0% had 11–15 years, and 8.6% had more than 15 years. In terms of years of experience in the present ward, 21.6% had less than one year; 56.4% had worked 1–5 years, and 2.4% had more than 15 years.

Table 1 presents a description of the nurses’ work environment, perceived person-centeredness, and nurses’ rated quality and patient safety in relation to their demographic characteristics. Higher rates of quality were among nurses more than 35 years old (3.91 ± 0.49, *p* < 0.05) and nurses with more than 15 years of total experience (3.90 ± 0.51, *p* < 0.05). Moreover, higher rates of quality of care (3.81 ± 0.55, *p* < 0.01) and a higher perception of person-centered care (3.69 ± 0.38, *p* < 0.05) were associated with being married. In comparison, Malay nurses perceived a better work environment (3.62 ± 0.43, *p* < 0.01), and rated higher quality (3.79 ± 0.57, *p* < 0.01) and safer care (3.66 ± 0.58, *p* < 0.01) than Chinese nurses.

### 3.3. Practice Characteristics of Nurses

Table 2 shows the practice characteristics of nurses who participated in the study. More than two-thirds of participants were working in large-sized (72.2%), teaching (60.9), and non-accredited (72.0%) hospitals. There were no significant differences in the perceived work environment, person-centeredness, and nurses’ rated quality and patient safety in relation to hospital size, teaching, and accreditation status.

More than two-thirds of the nurses worked 7-h (47.7%) and 8-h (17.5%) shifts across the sample. Nurses in 7-h (3.62 ± 0.43, *p* < 0.01) and 8-h (3.63 ± 0.41, *p* < 0.01) shifts reported a higher perceived work environment compared to those working a 12-h shift (3.46 ± 0.41, *p* < 0.01). Furthermore, higher perceptions of person-centered care were reported among nurses in 7-h (3.70 ± 0.39, *p* < 0.01) and 8-h (3.71 ± 0.42, *p* < 0.01) shifts compared to those in 12-h shifts (3.55 ± 0.35, *p* < 0.01).

Interestingly, more than one-third of the nurses cared for more than 15 patients (37.0%) in their last work shift. Further, around one-quarter of nurses cared for 11–15 patients (24.0%), which indicated a high patient-to-nurse ratio that might jeopardize care outcomes. Nurses assigned more than 15 patients were less likely to report a favorable work environment (3.53 ± 0.41, *p* < 0.05), perceived lower person-centered care (3.61 ± 0.36, *p* < 0.01), and rated lower patient safety (3.54 ± 0.62, *p* < 0.05) compared to those assigned less than five patients in their last working shift. Similarly, nurses assigned 11–15 patients (3.61 ± 0.38, *p* < 0.01) perceived lower person-centeredness compared to those assigned less than five patients (3.80 ± 0.34, *p* < 0.01). Thus, nurses caring for more than 11 patients were less likely to integrate patient preferences.

### 3.4. The Direct Effect of the Nurse Work Environment

The c′ paths of the Hayes Macro regression models explored the direct effect of nurse work environment subscales on perceived quality and patient safety, controlling the effect of nurse demographics and practice characteristics with the presence of a mediator, as shown in Table 3 and Table 4. The results indicated that nurse participation in hospital affairs significantly influenced the quality of care (t = 9.81, *p* < 0.001); nurse foundation for quality (t = 6.33, *p* < 0.001); nurse manager’s ability and leadership support (t = 5.70, *p* < 0.001); nurse-physician relationship (t = 6.64, *p* < 0.001). Further, nurse participation in hospital affairs (t = 6.38, *p* < 0.001), nurse manager’s ability and leadership support (t = 3.10, *p* < 0.01), and the nurse-physician relationship significantly influenced patient safety (t = 4.58, *p* < 0.001).

Greater participation in hospital affairs was related to 35% and 26% for improved quality and patient safety, respectively. The ability of a nurse manager and leadership support were associated with a 9% and 11% improvement in quality and patient safety, respectively. Further, a strong nurse-physician relationship was significantly associated with a 26% and 20% improvement in quality and patient safety, respectively. Being trained and engaging nurses with quality development programs was associated with a 29% improvement in quality, while it was not associated with patient safety. The study mediation analysis explains this inconsistency.

### 3.5. Mediation Effect of Person-Centeredness

To better understand how the nurse practice environment affects quality and patient safety, the mediation effect of person-centered care, between the relationship of hospital nurses’ work environment dimensions and nurses’ perceived quality and patient safety, was explored. Four models are shown for each outcome in Table 3 and Table 4. These models help in understanding the role of maintaining a healthy work environment in supporting person-centeredness and improving quality and patient safety.

The results revealed that the confidence interval of the indirect effect of nurse participation in hospital affairs (95% CI = 0.10 to 0.20), nurse foundation for quality of care (95% CI = 0.16 to 0.30), nurse manager’s ability and leadership support (95% CI = 0.08 to 0.18), and nurse-physician relationship (95% CI = 0.12 to 0.25) do not include zero. Therefore, person-centeredness mediates the relationship of nurse work environment dimensions on the quality of care. Further, concerning patient safety, the data analysis results revealed a significant mediation effect of person-centered care as the confidence interval of the indirect effect of nurse participation in hospital affairs (95% CI = 0.09 to 0.19), nurse foundation for quality of care (95% CI = 0.17 to 0.32), nurse manager’s ability and leadership support (95% CI = 0.07 to 0.16), and nurse-physician relationship (95% CI = 0.10 to 0.23) do not include zero. The direct effect of c′ paths of the relationships was significant; thus, partial mediation occurred. The direct effect of the nurse foundation, for quality of care on patient safety, was no longer significant with the presence of a mediator, indicating that full mediation has occurred.

The signs of a x b x c′ baths are positive, indicating that the mediator is a complementary mediator [50]. Nurses in a healthy work environment were more likely to integrate patient preferences, which, in turn, improve quality and patient safety. The results revealed at least one indirect effect different from zero [48]. This finding supports the hypothesis that person-centeredness mediates the effect of the work environment on quality and patient safety. Thus, the work environment indirectly affects quality and patient safety through patient-centeredness.

The results indicated the greater participation in hospital affairs was associated with a 27% improvement in person-centered care, which, in turn, indirectly improves the quality of care by 15% and patient safety by 13%. Being a trained and engaged nurse, with quality development programs, was associated with a 41% improvement in person-centered care, which, in turn, led to an improved quality of care by 22% and patient safety by 24%. Nurse Manager’s ability and leadership support was associated with a 19% improvement in person-centered care, which, in turn, led to an improvement of quality of care by 12% and patient safety by 11%. Further, a strong nurse-physician relationship was significantly associated with a 31% improvement in person-centered care, which, in turn, indirectly improved the quality of care by 18% and patient safety by 16%. The R^2^ values indicate that the study models, of the mediation effect of person-centeredness, explained about one-third of the variances of nurse-rated quality of care and one-quarter of the variances of nurse-rated patient safety at the *p* < 0.001 level of significance.

## 4. Discussion

This study provides evidence from the healthcare system in Malaysia, specifically from the private sector, which had complained about a high number of adverse events. The study results were supported and contributed to the Donabedian theory and SEIPS model [3,26]. Hence, the study model provided insights into providing quality and safe patient care and its associations with the structural and process indicators. The results revealed that several factors were associated with enhancing quality or mitigating patient harm. Furthermore, the results provided an insight into the role of a healthy work environment in supporting person-centeredness for improving quality and safety in the private hospitals in Malaysia. This expands the SEIPS model by focusing on the patient journey in a work system [2] and shifting to the human-centered system, considering the needs of patients and providers simultaneously. Therefore, focusing on the patient and providers’ journey in a work system.

The results presented the associations of nurses’ demographics and practice characteristics in relation to the nurse work environment, person-centeredness, quality, and patient safety. Senior nurses, in terms of age and years of experience, perceived a high quality of care. The fact that they have high patient responsibilities and more substantial experience could explain that they have greater work expectations and perceived higher quality of care [53]. Furthermore, ethnic Malay nurses perceived a more favorable work environment, and they rated quality and patient safety higher than Chinese nurses. However, decreasing disparities between ethnic groups is challenging [54]. These results provide dual considerations to be addressed in future research. First is an in-depth understanding of these associations to decrease disparities between ethnic groups. Second, patient-nurse relationships and treatment of minorities are critical, and future research should address them.

In support of previous studies, the practice characteristics of this study indicated that nurses with a high workload, in terms of duty length and patient ratio, perceived a less preferable practice environment, were less likely to integrate patient preferences, and more likely to come across adverse events in their respective units. Previous studies have indicated that a larger number of patients per nurse and working longer shifts were negatively associated with practice outcomes [20,55,56].

The Hayes Macro regression models provide meaningful results that can afford insights into the importance of the nurse work environment in enhancing person-centered care and, consequently, improving quality and patient safety. The first result indicated that nurse participation in hospital affairs indirectly affects the quality and patient safety through person-centeredness. This means that nurses with greater participation in hospital affairs have a higher degree of person-centeredness, which, in turn, improves both quality and patient safety.

Previous studies support these findings. Per the American Association of Critical-Care Nurses (AACN) report (2005), ensuring effective staff participation, as well as patient and family education, is required to improve quality and patient safety. Nurses with high participation and involvement have high practice and clinical outcomes [57,58]. Similarly, in the Malaysian healthcare sector, employee involvement, and participation are important factors to optimize care outcomes [59]. Furthermore, increasing nurses’ participation and enhancing their job engagement would reduce their physical and emotional exhaustion [60]. Therefore, they will be more likely to integrate patient preferences in their workplace. Hence, if nurses address the needs and interests of patients, this focus helps prevent patient harm and improve the quality of care [30].

The second result indicated that person-centeredness significantly mediates the effect of a nurse’s foundation for the quality of care on both the quality and patient safety. Simultaneously, the c′ path of the effect of a nurse’s foundation for the quality of care indicates a significant and positive effect on the quality of care and an insignificant effect on patient safety. This explains the inconsistency in previous studies. The nurse work environment dimensions were inconsistently related to care outcomes [21,22], and a mediator variable is required to interpret these associations.

According to the AHRQ, nurse managers should engage more nurses in quality improvement programs, continuous education, and training for improving quality and patient safety [61]. For instance, nurses involved in reducing medication error programs were spending a longer time for medication preparation and patients’ orientation [61], which helped improve care outcomes. Similarly, an interventional study in 15 wards in Malaysian hospitals found that nursing education and training were effective tools in improving the safety climate [62]. These findings support our assumption that a nurse foundation for quality of care, as a structural factor, affects care outcomes through nurse perceived person-centeredness as a process factor. Therefore, implementing a quality improvement program, upgrading equipment, and extensive training can improve the structure, process, and outcome quality. Hence, a nurse foundation for quality of care reduces nurse burnout and emotional distress [15]. Thus, nurses are more likely to integrate patient preferences, which, in turn, help with providing quality and safe patient care.

The third result indicated that person-centeredness mediates the effect of a nurse manager’s ability, leadership, and support on both quality and patient safety. Nurses with a skilled and supportive leader have a higher degree of person-centeredness to integrate patient preferences, which, in turn, improves both quality and patient safety. Previous literature has reported that trained and skilled leaders are required to improve the quality and patient safety [63]; enhance teamwork and person-centeredness [64]. According to the AHRQ, skilled leaders, effective decision-making, and collaboration are all required to sustain a healthy work environment. Furthermore, a comparative correlational survey in England and Malaysia found that Malaysian nurses were more obliged to their managers [65]. Therefore, in addition to having obligated nurses to their managers, safety organizational culture requires evidence-based leaders, having the ability to develop teamwork and involving nurses, to be more person-centered to improve quality and prevent patient harm [64]. Thus, this shows the importance of the nurse manager’s ability, leadership, and support in enhancing person-centered care, which, in turn, improves the outcomes of care.

The fourth result indicated that the nurse-physician relationship indirectly affects quality and patient safety through person-centeredness. Nurses with a strong relationship with collegial physicians have a higher degree of person-centeredness, which, in turn, improves both quality and patient safety. Previous studies support these findings. Hence, high nurse-physician collaboration reduces adverse events and promotes safety [66,67]. An interdisciplinary team with an excellent nurse-physician relationship helps sustain care outcomes [60]. The fact that nurses and physicians substitute for each other and complement each other’s roles, leading to decreased workloads, helps explain this [68]. Thus, nurses on good terms with the physicians spend more time with patients and provide more person-centered care, improving both the quality and patient safety. Effective communication among the multidisciplinary teams and periodical meetings of professionals is recommended for the outcome optimization.

### Limitations and Future Directions

The study was completed in 2015; the data reported, herein, were dated. This was a result of co-authors passing away since the manuscript was completed. The surviving authors updated the references and the theories supporting the study model. Because the study variables are interpersonal interactions in a work system, they are less likely to be affected by time. Furthermore, this study was a cross-sectional survey in Malaysian private hospitals at one point in time. Therefore, it is difficult to establish the causality between the study variables and generalize the study results. The study design limited the ability to assert a causal relationship between the nurse work environment, perceived person-centeredness, and nurse reported quality and patient safety. In addition, the study sample limited the ability to generalize the results, as it is conducted in Malaysia, and data collected from nurses in private hospitals represents 5.7% of the Malaysian private hospitals and 3.0% of total nurses in the private hospitals in Malaysia. However, as the quantitative approach suggests that generalization is possible, the findings in this report can be referred to as reliable, taking into consideration the randomness process of data collection that offers a reliable sample to represent the population. The multi-stage random sampling method ensures all nurses have an equal chance to be included in the study, thus makes the findings presentable.

Furthermore, important potential variables, such as nurse burnout, stress, fatigue, and nurse reported intention to leave were not explored and should be included in future work to understand the mediation role of person-centered care better. Additionally, the direct effect c′ paths of nurse reported quality and safety were significant, indicating that other mediators could be used for future research [50]. The study mediator had a positive impact on nurse reported quality and safety. Therefore, future research must include mediators such as nursing burnout, workaround, and staffing inadequacy with negative signs to understand the study model better. Finally, lacking data of the actual outcome quality, such as reported events and mortality rates, the data relied on nurse reported quality and patient safety. However, these measures were used widely and validated internationally [34,37,69].

## 5. Conclusions and Implications

The study provides new insights into nursing research and practice; it is the first study of its kind that adds to the increasing literature concerning the mediation role of person-centeredness between the effect of practice work environment on quality and patient safety. The findings provide significant promises of the importance of maintaining a healthy work environment in the provision of high person-centered care, improving the quality of care, and reducing patient harm. Nurses with a high workload were less likely to report a positive work environment, less likely to integrate patient preferences, and more likely to encounter adverse events. Policymakers need to re-evaluate nurses’ assignments and staffing levels in Malaysian private hospitals to maintain staffing adequacy and reduce their workload.

Moreover, the findings underscore that maintaining a work practice environment that focuses on task quality, in which nurse managers focus on the nurses’ performance by engaging them in hospital affairs, provide training in quality development programs, leadership support, and maintain collegial relationships help in improving person-centered care. This, in turn, improves the quality of care and patient safety. Therefore, nurse managers must balance task-oriented and patient-oriented leadership to simultaneously improve staff and patient care outcomes.

The function of person-centeredness, in the study, was to complement the work environment’s impact on care outcomes. Therefore, a nurse manager should sustain a healthy work environment by enhancing nurses’ engagement and participation in hospital affairs to increase person-centeredness, which, in turn, improves the outcomes of care. They should involve nurses in new policies and procedures; encourage them to adopt the evidence-based practices of the current clinical research into their clinical practices. Furthermore, nurse managers should maintain trained and educated nurses for improving quality and patient safety for strengthening a healthy work environment of nurses. Therefore, they should channel resources to develop programs of support for nurses, training and learning to improve the skills of nurses and potential future nurse leaders, maintaining the communication and teamwork among care providers, and instilling a culture of person-centeredness to design a human-centered system for improving both quality and patient safety.

## Figures and Tables

**Figure 1 healthcare-09-01578-f001:**
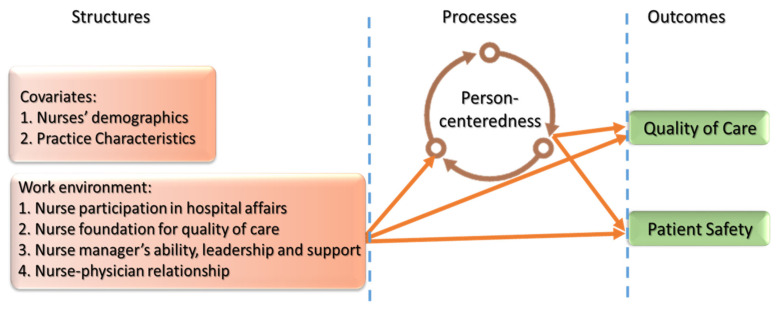
Research framework.

**Table 1 healthcare-09-01578-t001:** Nurses’ Demographic Data and Differences between Nurses in the Perceived Practice Environment, Person-centeredness, Quality, and Patient Safety.

Demographic Characteristics	Categories (*n*, %)	Work Environment Mean (SD)	Person-Centeredness Mean (SD)	Quality of Care Mean (SD)	Patient Safety Mean (SD)
Nationality	Malaysian (577, 99.0) Non-Malaysian (6, 1.0)	3.58 (0.43) 3.47 (0.56)	3.65 (0.39) 3.85 (0.39)	3.74 (0.56) 3.37 (0.64)	3.59 (0.59) 3.19 (0.35)
Age	<25 years (196, 33.6) 25–30 years (255, 43.7) 31–35 years (71, 12.2) >35 years (61, 10.5)	3.75 (0.45) 3.56 (0.43) 3.63 (0.39) 3.62 (0.37)	3.65 (0.43) 3.62 (0.38) 3.67 (0.36) 3.75 (0.35)	3.70 (0.59) 3.69 (0.57) 3.81 (0.54) 3.91 (0.49) *	3.63 (0.56) 3.57 (0.59) 3.50 (0.61) 3.61 (0.63)
Gender	Male (14, 2.4) Female (569, 97.6)	3.70 (0.50) 3.58 (0.43)	3.68 (0.39) 3.65 (0.39)	3.83 (0.58) 3.73 (0.57)	3.69 (0.53) 3.58 (0.59)
Marital status	Married (264, 45.3) Single (306, 52.5) Others (13, 2.2)	3.61 (0.42) 3.55 (0.43) 3.49 (0.52)	3.69 (0.38) *^a^ 3.63 (0.39) 3.44 (0.39)	3.81 (0.55) **^a^ 3.67 (0.57) 3.63 (0.64)	3.61 (0.60) 3.57 (0.57) 3.44 (0.70)
Race	Malay (350, 60.0) Chinese (126, 21.6) Indian (83, 14.2) Others (24, 4.1)	3.62 (0.43) **^b^ 3.46 (0.40) 3.57 (0.41) 3.61 (0.48)	3.66 (0.39) 3.59 (0.40) 3.73 (0.36) 3.64 (0.36)	3.79 (0.57) **^b^ 3.59 (0.61) 3.71 (0.47) 3.66 (0.45)	3.66 (0.58) **^b^ 3.44 (0.54) 3.56 (0.54) 3.40 (0.54)
Education level	Bachelor’s (60, 10.3) Diploma (493, 84.6) Others (30, 5.1)	3.49 (0.49) 3.60 (0.42) 3.48 (0.36)	3.64 (0.34) 3.65 (0.40) 3.75 (0.42)	3.67 (0.63) 3.73 (0.56) 3.85 (0.45)	3.54 (0.62) 3.59 (0.59) 3.56 (0.60)
Years of experience	0–5 years (384, 65.9) 6–10 years (108, 18.5) 11–15 years (41, 7.0) >15 years (50, 8.6)	3.56 (0.43) 3.61 (0.46) 3.66 (0.34) 3.61 (0.38)	3.63 (0.40) 3.68 (0.38) 3.75 (0.37) 3.72 (0.37)	3.69 (0.57) 3.77 (0.55) 3.85 (0.59) 3.90 (0.51) *	3.61 (0.56) 3.55 (0.64) 3.49 (0.67) 3.60 (0.64)
Experience in the current ward	<1 year (126, 21.6) 1–5 years (329, 56.4) 6–10 years (88, 15.1) 11–15 years (26, 4.5) >15 years (14, 2.4)	3.67 (0.43) 3.54 (0.42) 3.59 (0.46) 3.66 (0.30) 3.50 (0.33)	3.69 (0.39) 3.62 (0.40) 3.67 (0.37) 3.76 (0.41) 3.84 (0.31)	3.74 (0.59) 3.69 (0.56) 3.84 (0.53) 3.91 (0.61) 3.74 (0.47)	3.68 (0.59) 3.58 (0.56) 3.53 (0.64) 3.47 (0.62) 3.41 (0.76)

Notes: *: *p* < 0.05, **: *p* < 0.01. a: Married nurses had a significantly higher mean value than single nurses. b: Malay nurses had a significantly higher mean value than Chinese.

**Table 2 healthcare-09-01578-t002:** Nurses’ Practice Characteristics and Differences between Nurses in the Perceived Practice Environment, Person-centeredness, Quality, and Patient Safety.

Practice Characteristics	Categories (*n*, %)	Work Environment Mean (SD)	Person-Centeredness Mean (SD)	Quality of Care Mean (SD)	Patient Safety Mean (SD)
Hospital size; beds	Small; <100 (66, 11.3) Medium; 100–199 (96, 16.5) Large; >200 (421, 72.2)	3.57 (0.38) 3.67 (0.41) 3.56 (0.44)	3.60 (0.35) 3.67 (0.33) 3.66 (0.41)	3.63 (0.51) 3.83 (0.55) 3.73 (0.57)	3.62 (0.57) 3.67 (0.56) 3.56 (0.60)
Teaching status	Teaching (355, 60.9) Non-teaching (228, 39.1)	3.55 (0.45) 3.62 (0.39)	3.64 (0.41) 3.68 (0.36)	3.73 (0.58) 3.73 (0.55)	3.56 (0.60) 3.63 (0.56)
Accreditation status	Accredited (162, 28.0) Non-accredited (420, 72.0)	3.56 (0.40) 3.59 (0.44)	3.65 (0.37) 3.65 (0.40)	3.70 (0.53) 3.75 (0.58)	3.60 (0.54) 3.58 (0.60)
Total working hours in the last working shift	7 h (278, 47.7) 8 h (102, 17.5) 10 h (98, 16.8) 12 h (93, 16.0) Others (12, 2.1)	3.62 (0.43) **^a^ 3.63 (0.41) ** ^a^ 3.52 (0.43) 3.46 (0.41) 3.50 (0.43)	3.70 (0.39) **^a^ 3.71 (0.42) **^a^ 3.57 (0.37) 3.55 (0.35) 3.65 (0.40)	3.74 (0.58) 3.80 (0.53) 3.63 (0.61) 3.73 (0.50) 3.77 (0.61)	3.63 (0.60) 3.61 (0.58) 3.47 (0.59) 3.52 (0.52) 3.49 (0.76)
Number of patient under your care	<5 patients (45, 7.7) 5–10 patients (182, 31.2) 11–15 patients (140, 24.0) >15 patients (216, 37.0)	3.75 (0.40) *^b^ 3.60 (0.44) 3.57 (0.43) 3.53 (0.41)	3.80 (0.34) **^b^ 3.70 (0.44) 3.61 (0.38) 3.61 (0.36)	3.90 (0.56) 3.78 (0.61) 3.70 (0.51) 3.68 (0.55)	3.80 (0.57) *^b^ 3.62 (0.56) 3.55 (0.57) 3.54 (0.62)

Notes: *: *p* < 0.05, **: *p* < 0.01. a: Nurses working 7-h and 8-h shifts had a significantly higher mean value than nurses working 12-h shifts. b: Nurses with less than 5 patients had a significantly higher mean value.

**Table 3 healthcare-09-01578-t003:** The Mediation Effect of Person-Centeredness on the Quality of Care.

Practice Environment Dimensions	Value	Paths					Model Summary		
		a path	b path	c path	c′ path	a x b path (LCI, UCI)	R^2^	F	*p*
Nurse participation in hospital affairs	Coefficient St. Error t *p*	0.27 *** 0.03 10.23 0.000	0.54 *** 0.05 10.39 0.000	0.50 *** 0.04 13.87 0.000	0.35 *** 0.04 9.81 0.000	0.15 (0.10, 0.20)	0.39	37.26	0.000
Nurse foundation for quality of care	Coefficient St. Error t *p*	0.41 *** 0.03 13.92 0.0000	0.55 *** 0.06 9.63 0.000	0.51 *** 0.04 12.00 0.000	0.29 *** 0.05 6.33 0.000	0.22 (0.16, 0.30)	0.34	29.33	0.000
Nurse manager’s ability, leadership, and support	Coefficient St. Error t *p*	0.19 *** 0.02 7.68 0.000	0.65 *** 0.05 12.29 0.000	0.31 *** 0.04 8.84 0.000	0.19 *** 0.03 5.70 0.000	0.12 (0.08, 0.18)	0.33	28.26	0.000
Nurse-physician relationship	Coefficient St. Error t *p*	0.31 *** 0.03 11.76 0.000	0.58 *** 0.06 10.42 0.000	0.44 *** 0.04 11.48 0.000	0.26 *** 0.04 6.64 0.000	0.18 (0.12, 0.25)	0.34	29.89	0.000

Notes: ***: *p* < 0.001. a path: the effect of work environment dimensions on person-centeredness. b path: direct effect of person-centeredness on the quality of care. c path: total effect of work environment dimensions on the quality of care. c′ path: direct effect of work environment dimensions on the quality of care. a x b path: indirect effect of work environment dimensions on the quality of care through person-centeredness. LCI, UCI: lower and upper confidence interval of the indirect effect.

**Table 4 healthcare-09-01578-t004:** The Mediation Effect of Person-centeredness on Patient Safety.

Practice Environment Dimensions	Value	Paths					Model Summary		
		a path	b path	c path	c′ path	a x b path (LCI, UCI)	R^2^	F	*p*
Nurse participation in hospital affairs	Coefficient St. Error t *p*	0.27 *** 0.03 10.23 0.000	0.50 *** 0.06 8.35 0.000	0.39 *** 0.04 9.93 0.000	0.26 *** 0.04 6.38 0.000	0.13 (0.09, 0.19)	0.27	20.67	0.000
Nurse foundation for quality of care	Coefficient St. Error t *p*	0.41 *** 0.03 13.92 0.0000	0.58 *** 0.07 8.92 0.000	0.34 *** 0.05 6.97 0.000	0.10 0.05 1.94 0.053	0.24 (0.17, 0.32)	0.22	15.97	0.000
Nurse manager’s ability, leadership, and support	Coefficient St. Error t *p*	0.19 *** 0.02 7.68 0.000	0.59 *** 0.06 10.00 0.000	0.22 *** 0.04 5.97 0.000	0.11 ** 0.04 3.10 0.002	0.11 (0.07, 0.16)	0.23	16.71	0.000
Nurse-physician relationship	Coefficient St. Error t *p*	0.31 *** 0.03 11.76 0.000	0.52 *** 0.06 8.40 0.000	0.36 *** 0.04 8.71 0.000	0.20 *** 0.04 4.58 0.000	0.16 (0.10, 0.23)	0.24	18.16	0.000

Notes: **: *p* < 0.01, ***: *p* < 0.001. a path: the effect of work environment dimensions on person-centeredness. b path: direct effect of person-centeredness on patient safety. c path: total effect of work environment dimensions on the quality of care. c′ path: direct effect of work environment dimensions on patient safety. a x b path: indirect effect of work environment dimensions on patient safety through person-centeredness. LCI, UCI: lower and upper confidence interval of the indirect effect.

## Data Availability

The datasets used and/or analyzed during the current study are available from the corresponding author (MJ) on reasonable request.

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
