# Peer review of "Effect of Practice Environment on Nurse Reported Quality and Patient Safety: The Mediation Role of Person-Centeredness"

_healthcare, 2021, doi:10.3390/healthcare9111578_

Round 1

Reviewer 1 Report

Mu'taman investigated the effect of practice environment on nurse reported quality and patient safety in Malaysia. The topic is important and interesting as numerous studies reported that the quality of patient care provided by nurses is significantly associated with patient outcome. My major concern is both quality of care and patient safety are determined based on nurses’ judgement in this study. However, as we know, medical staffs are not always willing to report patient harm or hospital-acquired conditions. I think this compromised this study. Additionally, this study should be reviewed by a statistician as well.

Author Response

Dear Sir,

This covering letter is to respond to the reviewers comments of our manuscript (Manuscript ID: healthcare-1438887)  for consideration for publication in the journal “Healthcare”.

Thank you for the comments made by all reviewers to optimize our manuscript entitled ““[Effect of Practice Environment on Nurse Reported Quality and Patient Safety: The Mediation Role of Person-Centeredness]”. Whilst improving the paper according to the suggestion, I would ask please to find the attached file and see our responses to the comments.

We confirm that this work is non-funded and original and is available as preprint in the research square (https://www.researchsquare.com/article/rs-797451/v1), and it is not under consideration for publication elsewhere. The Manuscript reviewed by native English proofreader.

This also to confirm that each author of this work has substantial contributions, critically revised the content and approved the final version of the manuscript. Attached with this letter the CRediT author statement.

In this paper, insights for nursing and nurse managers were provided. Please address all correspondence concerning this manuscript to me at [mutaman.jarrar@yahoo.com, mkjarrar@iau.edu.sa].

Thank you for reviewing our manuscript.

Sincerely,

Mu’taman Jarrar

Reviewer 2 Report

I would like to congratulate the authors for their interest in researching in this field, however, the work presented presents some deficiencies.

a) Abstract does not provide information for each part of the work, so it would not comply with the fundamental information of a research summary.

More specifically, while you adequately describe the data acquisition process, you should briefly explain the methodology used to process the study data.

b) Authors adequately define the data acquisition process, although they do not describe in detail the survey used.

This point is important to verify the adequate development of the research.

I suggest that the authors include the description in Chapter 2.

c) The research focuses the study population on a restricted geographic area. Are the results extrapolable to other regions? This point should be explicitly stated in the article to clarify the scope of the results obtained and their concrete applicability.

I hope that these changes will help to improve your article and make it a document of great scientific interest.

Author Response

(The authors gave the same response as above.)

Reviewer 3 Report

The authors investigated the impact of practice environment on nurse reported quality and patient safety. 

The introduction is clear and straight forward. Methods and statistics are adequate. Ethical approval is mentioned.

Results are significant and convincing. Four major results are discussed comprehensively and limitations of the study as well.     

Author Response

(The authors gave the same response as above.)

Round 2

Reviewer 1 Report

My concerns have been addressed and I don't have more questions.